# CONVERGENCE RATE OF SIGN STOCHASTIC GRADIENT DESCENT FOR NON-CONVEX FUNCTIONS

## ABSTRACT

The sign stochastic gradient descent method (signSGD) utilises only the sign of the stochastic gradient in its updates. For deep networks, this one-bit quantisation has surprisingly little impact on convergence speed or generalisation performance compared to SGD. Since signSGD is effectively compressing the gradients, it is very relevant for distributed optimisation where gradients need to be aggregated from different processors. What's more, signSGD has close connections to common deep learning algorithms like RMSprop and Adam. We study the base theoretical properties of this simple yet powerful algorithm. For the first time, we establish convergence rates for signSGD on general non-convex functions under transparent conditions. We show that the rate of signSGD to reach first-order critical points matches that of SGD in terms of number of stochastic gradient calls, but loses out by roughly a linear factor in the dimension for general non-convex functions. We carry out simple experiments to explore the behaviour of sign gradient descent (without the stochasticity) close to saddle points and show that it can help to completely avoid certain kinds of saddle points without using either stochasticity or curvature information.

## 1 INTRODUCTION

Deep neural network training takes place in an error landscape that is high-dimensional, non-convex and stochastic. In practice, simple optimization techniques perform surprisingly well but have very limited theoretical understanding. While stochastic gradient descent (SGD) is widely used, algorithms like Adam (Kingma & Ba, 2015), RMSprop (Tieleman & Hinton, 2012) and Rprop (Riedmiller & Braun, 1993) are also popular. These latter algorithms involve component-wise rescaling of gradients, and so bear closer relation to signSGD than SGD. Currently, convergence rates have only been derived for close variants of SGD for general non-convex functions, and indeed the Adam paper gives convex theory.

Recently, another class of optimization algorithms has emerged which also pays attention to the resource requirements for training, in addition to obtaining good performance. Primarily, they focus on reducing costs for communicating gradients across different machines in a distributed training environment (Seide et al., 2014; Strom, 2015; Li et al., 2016; Alistarh et al., 2017; Wen et al., 2017). Often, the techniques involve quantizing the stochastic gradients at radically low numerical precision. Empirically, it was demonstrated that one can get away with using only one-bit per dimension without losing much accuracy (Seide et al., 2014; Strom, 2015). The theoretical properties of these approaches are however not well-understood. In particular, it was not known until now how quickly signSGD (the simplest incarnation of one-bit SGD) converges or even whether it converges at all to the neighborhood of a meaningful solution.

**Our contribution**: we supply the non-convex rate of convergence to first order critical points for signSGD. The algorithm updates parameter vector $x_k$ according to

$$x_{k+1} = x_k - \delta_k \text{sign}(\bar{g}_k) \tag{1}$$

where $\bar{g}_k$ is the mini-batch stochastic gradient and $\delta_k$ is the learning rate. We show that for non-convex problems, signSGD entertains convergence rates as good as SGD, up to a linear factor in the dimension. Our statements impose a particular learning rate and mini-batch schedule.

Ours is the first work to provide non-convex convergence rates for a *biased* quantisation procedure as far as we know, and therefore does not require the randomisation that other gradient quantisation algorithms need to ensure unbiasedness. The technical challenge we overcome is in showing how to carry the stochasticity in the gradient through the sign non-linearity of the algorithm in a controlled-fashion.

Whilst our analysis is for first order critical points, we experimentally test the performance of sign gradient descent without stochasticity (signGD) around saddle points. We removed stochasticity in order to investigate whether signGD has an inherent ability to escape saddle points, which would suggest superiority over gradient descent (GD) which can take exponential time to escape saddle points if it gets too close to them (Du et al., 2017).

In our work we make three assumptions. Informally, we assume that the objective function is lower-bounded, smooth, and that each component of the stochastic gradient has bounded variance. These assumptions are very general and hold for a much wider class of functions than just the ones encountered in deep learning.

**Outline of paper**: in Sections 3, 4 and 5 we give non-convex theory of signSGD. In Section 6 we experimentally test the ability of the signGD (without the S) to escape saddle points. And in Section 7 we pit signSGD against SGD and Adam on CIFAR-10.

## 2 RELATED WORK

**Deep learning:** the prototypical optimisation algorithm for neural networks is stochastic gradient descent (SGD)—see Algorithm 2. The deep learning community has discovered many practical tweaks to ease the training of large neural network models. In Rprop (Riedmiller & Braun, 1993) each weight update ignores the magnitude of the gradient and pays attention only to the sign, bringing it close to signSGD. It differs in that the learning rate for each component is modified depending on the consistency of the sign of consecutive steps. RMSprop (Tieleman & Hinton, 2012) is Rprop adapted for the minibatch setting—instead of dividing each component of the gradient by its magnitude, the authors estimate the rescaling factor as an average over recent iterates. Adam (Kingma & Ba, 2015) is RMSprop with momentum, meaning both gradient and gradient rescaling factors are estimated as bias-corrected averages over iterates. Indeed switching off the averaging in Adam yields signSGD. These algorithms have been applied to a breadth of interesting practical problems, e.g. (Xu et al., 2015; Gregor et al., 2015).

In an effort to characterise the typical deep learning error landscape, Dauphin et al. (2014) frame the primary obstacle to neural network training as the proliferation of saddle points in high dimensional objectives. Practitioners challenge this view, suggesting that saddle points may be seldom encountered at least in retrospectively successful applications of deep learning (Goodfellow et al., 2015).

**Optimisation theory:** in convex optimisation there is a natural notion of success—rate of convergence to the global minimum $x^*$. Convex optimisation is eased by the fact that *local information* in the gradient provides *global information* about the direction towards the minimum, i.e. $\nabla f(x)$ tells you information about $x^* - x$.

In non-convex problems finding the global minimum is in general intractable, so theorists usually settle for measuring some restricted notion of success, such as rate of convergence to stationary points (e.g. Allen-Zhu (2017a)) or local minima (e.g. Nesterov & Polyak (2006)). Given the importance placed by Dauphin et al. (2014) upon evading saddle points, recent work considers the efficient use of noise (Jin et al., 2017; Levy, 2016) and curvature information (Allen-Zhu, 2017b) to escape saddle points and find local minima.

**Distributed machine learning:** whilst Rprop and Adam were proposed by asking how we can use gradient information to make better optimisation steps, another school asks how much information can we throw away from the gradient and still converge at all. Seide et al. (2014); Strom (2015) demonstrated empirically that one-bit quantisation can still give good performance whilst dramatically reducing gradient communication costs in distributed systems. Convergence properties of quantized stochastic gradient methods remain largely unknown. Alistarh et al. (2017) provide convergence rates for quantisation schemes that are unbiased estimators of the true gradient, and are

thus able to rely upon vanilla SGD convergence results. Wen et al. (2017) prove asymptotic convergence of a $\{-1, 0, 1\}$ ternary quantization scheme that also retains the unbiasedness of the stochastic gradient. Our proposed approach is different, in that we directly employ the sign gradient which is *biased*. This avoids the randomization needed for constructing an unbiased quantized estimate. To the best of our knowledge, the current work is the first to establish a convergence rate for a biased quantisation scheme, and our proof differs to that of vanilla SGD.

**Parallel work:** signSGD is related to both attempts to *improve* gradient descent like Rprop and Adam, and attempts to *damage it but not too badly* like quantised SGD. After submitting we became aware that Anonymous (2018) also made this link in a work submitted to the same conference. Our work gives non-convex theory of signSGD, whereas their work analyses Adam in greater depth, but only in the convex world.

## 3 ASSUMPTIONS

**Assumption 1** (The objective function is bounded below). *For all $x$ and some constant $f^*$, the objective function satisfies*

$$f(x) \geq f^* \tag{2}$$

Remark: this assumption applies to every practical objective function that we are aware of.

**Assumption 2** (The objective function is L-Lipschitz smooth). *Let $g(x)$ denote the gradient of the objective $f(.)$ evaluated at point $x$. Then for every $y$ we assume that*

$$\left| f(y) - \left[ f(x) + g(x)^T (y - x) \right] \right| \leq \frac{L}{2} \|y - x\|_2^2 \tag{3}$$

Remark: this assumption allows us to measure the error in trusting the local linearisation of our objective, which will be useful for bounding the error in a single step of the algorithm. For signSGD we can actually relax this assumption to only hold only for $y$ within a local neighbourhood of $x$, since signSGD takes steps of bounded size.

**Assumption 3** (Stochastic gradient oracle). *Upon receiving query $x$, the stochastic gradient oracle gives us an* independent *estimate $\hat{g}$ satisfying*

$$\mathbb{E}[\hat{g}(x)] = g(x), \qquad\qquad \text{Var}(\hat{g}(x)[i]) \leq \sigma^2 \; \forall i = 1, ..., d.$$

Remark: this assumption is standard for stochastic optimization, except that the variance upper bound is now stated for every dimension separately. A realization of the above oracle is to choose a data point uniformly at random, and to evaluate its gradient at point $x$. In the algorithm, we will be working with a minibatch of size $n_k$ in the $k^{th}$ iteration, and the corresponding minibatch stochastic gradient is modeled as the average of $n_k$ calls of the above stochastic gradient oracle at $x_k$. Therefore in this case the variance bound is squashed to $\sigma^2 / n_k$.

## 4 NON-CONVEX CONVERGENCE RATE OF SIGNSGD

Informally, our primary result says that if we run signSGD with the prescribed learning rate and mini-batch schedules, then after $N$ stochastic gradient evaluations, we should expect that *somewhere along the optimisation trajectory* will be a place with gradient 1-norm smaller than $O(N^{-0.25})$. This matches the non-convex SGD rate, insofar as they can be compared, and ignoring all (dimension-dependent!) constants.

Before we dive into the theorems, here's a refresher on our notation—deep breath—$g_k$ is the gradient at step $k$, $f^*$ is the lower bound on the objective function, $f_0$ is the initial value of the objective function, $d$ is the dimension of the space, $K$ is the total number of iterations, $N_K$ is the cumulative number of stochastic gradient calls at step $K$, $\sigma$ is the intrinsic variance-proxy for each component of the stochastic gradient, and finally $L$ is the maximum curvature (see Assumption 2).

---

**Algorithm 1** Sign stochastic gradient descent (signSGD)

---
1: **Inputs**: $x_0, K$ ▷ initial point and time budget
2: **for** $k \in [0, K-1]$ **do**
3:     $\delta_k \leftarrow \text{learningRate}(k)$
4:     $n_k \leftarrow \text{miniBatchSize}(k)$
5:     $\bar{g}_k \leftarrow \frac{1}{n_k} \sum_{i=1}^{n_k} \text{stochasticGradient}(x_k)$
6:     $x_{k+1} \leftarrow x_k - \delta_k \text{sign}(\bar{g}_k)$ ▷ the sign operation is element-wise

---

**Algorithm 2** Stochastic gradient descent

---
1: **Inputs**: $x_0, K$ ▷ initial point and time budget
2: **for** $k \in [0, K-1]$ **do**
3:     $\delta_k \leftarrow \text{learningRate}(k)$
4:     $n_k \leftarrow \text{miniBatchSize}(k)$
5:     $\hat{g}_k \leftarrow \frac{1}{n_k} \sum_{i=1}^{n_k} \text{stochasticGradient}(x_k)$
6:     $x_{k+1} \leftarrow x_k - \delta_k \bar{g}_k$

---

**Theorem 1** (Non-convex convergence rate of signSGD). *Apply Algorithm 1 under Assumptions 1, 2 and 3. Schedule the learning rate and mini-batch size as*

$$\delta_k = \frac{\delta}{\sqrt{k+1}} \qquad n_k = k+1 \qquad (4)$$

*Let $N_K$ be the cumulative number of stochastic gradient calls up to step $K$, i.e. $N_K = O(K^2)$ Then we have*

$$\mathbb{E}\left[\min_{0 \leq k \leq K-1} \|g_k\|_1\right]^2 \leq \frac{1}{\sqrt{N_{K-2}}}\left[\frac{f_0 - f_*}{\delta} + d(2 + \log(2N_{K-1}))(\sigma + \delta L)\right]^2 \qquad (5)$$

---

**Theorem 2** (Non-convex convergence rate of stochastic gradient descent). *Apply Algorithm 2 under Assumptions 1, 2 and 3. Schedule the learning rate and mini-batch size as*

$$\delta_k = \frac{\delta}{\sqrt{k+1}} \qquad n_k = 1 \qquad (6)$$

*Let $N_K$ be the cumulative number of stochastic gradient calls up to step $K$, i.e. $N_K = K$. Then we have that*

$$\mathbb{E}\left[\min_{0 \leq k \leq K-1} \|g_k\|_2^2\right] \leq \frac{1}{\sqrt{N_K}}\left[\frac{f_0 - f^*}{\delta\left(1 - \frac{\delta L}{2}\right)} + d(1 + \log N_K)\frac{\frac{\delta L}{2}}{1 - \frac{\delta L}{2}}\sigma^2\right] \qquad (7)$$

---

The proofs are deferred to Appendix B and here we sketch the intuition for Theorem 1. First consider the non-stochastic case: we know that if we take lots of steps for which the gradient is large, we will make lots of progress downhill. But since the objective function has a lower bound, it is impossible to keep taking large gradient steps downhill indefinitely, therefore increasing the number of steps requires that we must run into somewhere with small gradient.

To get a handle on this analytically, we must bound the per-step improvement in terms of the norm of the gradient. Assumption 2 allows us to do exactly this. Then we know that the sum of the per-step improvements over all steps must be smaller than the total possible improvement, and that gives us a bound on how large the minimum gradient that we see can be.

In the non-stochastic case, the obstacle to this process is curvature. Curvature means that if we take too large a step the gradient becomes unreliable, and we might move uphill instead of downhill. Since the step size in signSGD is set purely by the learning rate, this means we must anneal the learning rate if we wish to be sure to control the curvature-induced error and make good progress downhill. Stochasticity also poses a problem in signSGD. In regions where the gradient signal is

smaller than the noise, the noise is enough to flip the sign of the gradient. This is more severe than the additive noise in SGD, and so the batch size must be grown to control this effect.

You might expect that growing the batch size should lead to a worse convergence rate than SGD. This is forgetting that signSGD has an advantage in that it takes large steps even when the gradient is small. It turns out that this positive effect cancels out the fact that the batch size needs to grow, and the convergence rate ends up being the same as SGD.

For completeness, we also present the convergence rate for SGD derived under our assumptions. The proof is given in Appendix C. Note that this appears to be a classic result, although we are not sure of the earliest reference. Authors often hide the dimension dependence of the variance bound. SGD does not require an increasing batch size since the effect of the noise is second order in the learning rate, and therefore gets squashed as the learning rate decays. The rate ends up being the same in $N_K$ as signSGD because SGD makes slower progress when the gradient is small.

## 5 COMPARING THE CONVERGENCE RATE TO SGD

To make a clean comparison, let us set $\delta = \frac{1}{L}$ (as is often recommended) and hide all numerical constants in Theorems 1 and 2. Then for signSGD, we get

$$\mathbb{E}\Big[\min\|g_k\|_1\Big]^2 \sim \frac{1}{\sqrt{N}}\Big[L(f_0 - f^*) + d(\sigma + 1)\log N\Big]^2; \tag{8}$$

and for SGD we get

$$\mathbb{E}\Big[\min\|g_k\|_2^2\Big] \sim \frac{1}{\sqrt{N}}\Big[L(f_0 - f^*) + d\sigma^2 \log N\Big] \tag{9}$$

where $\sim$ denotes general scaling. What do these bounds mean? They say that after we have made a cumulative number of stochastic gradient evaluations $N$, that we should expect *somewhere along our trajectory* to have hit a point with gradient norm smaller than $N^{-\frac{1}{4}}$.

One important remark should be made. SignSGD more naturally deals with the one norm of the gradient vector, hence we had to square the bound to enable direct comparison with SGD. This means that the constant factor in signSGD is roughly worse by a square. Paying attention only to dimension, this looks like

$$\text{signSGD: } \mathbb{E}\Big[\min\|g_k\|_1\Big]^2 \sim \frac{d^2}{\sqrt{N}} \qquad \text{SGD: } \mathbb{E}\Big[\min\|g_k\|_2^2\Big] \sim \frac{d}{\sqrt{N}} \tag{10}$$

This defect in dimensionality should be expected in the bound, since signSGD almost never takes the direction of steepest descent, and the direction only gets worse as dimensionality grows. This raises the question, why do algorithms like Adam, which closely resemble signSGD, work well in practice?

Whilst answering this question fully is beyond the scope of this paper, we want to point out one important detail. Whilst the signSGD bound is worse by a factor $d$, it is also making a statement about the 1-norm of the gradient. Since the 1-norm of the gradient is always larger than the 2-norm, the signSGD bound is *stronger* in this respect. Indeed, if the gradient is distributed roughly uniformly across all dimensions, then the squared 1-norm is roughly $d$ times bigger than the squared 2-norm, i.e.

$$\|g_k\|_1^2 \sim d\|g_k\|_2^2$$

and in this limit both SGD and signSGD have a bound that scales as $\frac{d}{\sqrt{N}}$.

## 6 SWINGING BY SADDLE POINTS? AN EXPERIMENT

Seeing as our theoretical analysis only deals with convergence to stationary points, it does not address how signSGD might behave around saddle points. We wanted to investigate the naïve intuition that gradient rescaling should help flee saddle points—or in the words of Zeyuan Allen-Zhu—swing by them.

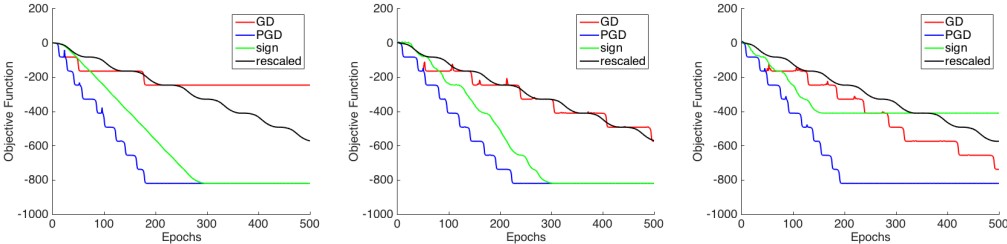

Figure 1: Descending the tube function of (Du et al., 2017). To reach the minimum, the algorithm must navigate a series of saddle points. Optimisers tested were gradient descent (GD), perturbed gradient descent (PGD) (Jin et al., 2017), sign gradient descent (signGD) and the rescaled gradient method (noiseless version of (Levy, 2016)). No learning rate tuning was attempted, so we suggest only focusing on the qualitative behaviour. *Left:* signGD appears not to 'see' the saddle points in the original objective function. *Middle:* after breaking the objective function's axis alignment by rotating it, the sign method's performance is still quantitatively different. Also the numerical error in our rotation operation appears to help unstick GD from the saddle points, illustrating the brittleness of Du et al. (2017)'s construction. *Right:* for some rotations, the sign method (with fixed learning rate and zero stochasticity) can get stuck in perfectly periodic orbits around saddle points.

For a testbed, the authors of (Du et al., 2017) kindly provided their 10-dimensional 'tube' function. The tube is a specially arranged gauntlet of saddle points, each with only one escape direction, that must be navigated in sequence before reaching the global minimum of the objective. The tube was designed to demonstrate how stochasticity can help escape saddles. Gradient descent takes much longer to navigate the tube than *perturbed gradient descent* of (Jin et al., 2017). It is interesting to ask, even empirically, whether the sign non-linearity in signSGD can also help escape saddle points efficiently. For this reason we strip out the stochasticity and pit the sign gradient descent method (signGD) against the tube function.

There are good reasons to expect that signGD might help escape saddles—for one, it takes large steps even when the gradient is small, which could drive the method away from regions of small gradient. For another, it is able to move in directions orthogonal to the gradient, which might help discover escape directions of the saddle. We phrase this as signGD's greater ability to *explore*.

Our experiments revealed that these intuitions sometimes hold out, but there are cases where they break down. In Figure 1, we compare the sign gradient method against gradient descent, perturbed gradient descent (Jin et al., 2017) and rescaled gradient descent $\left(x_{k+1} = x_k - \frac{g}{\|g\|_2}\right)$ which is a noiseless version of the algorithm considered in (Levy, 2016). No learning rate tuning was conducted, so we suggest paying attention to the qualitative behaviour rather than the ultimate convergence speed. The left hand plot pits the algorithms against the vanilla tube function. SignGD has very different qualitative behaviour to the other algorithms—it appears to make progress completely unimpeded by the saddles. We showed that this behaviour is partly due to the axis alignment of the tube function, since after randomly rotating the objective the behaviour changes (although it is still qualitatively different to the other algorithms).

One unexpected result was that for certain random rotations of the objective, signGD could get stuck at saddle points (see right panel in Figure 1). On closer inspection, we found that the algorithm was getting stuck in perfect periodic orbits around the saddle. Since the update is given by the learning rate multiplied by a binary vector, if the learning rate is constant it is perfectly possible for a sequence of updates to sum to zero. We expect that this behaviour relies on a remarkable structure in both the tube function and the algorithm. We hypothesise that for higher dimensional objectives and a non-fixed learning rate, this phenomenon might become extremely unlikely. This seems like a worthy direction of future research. Indeed we found empirically that introducing momentum into the update rule was enough to break the symmetry and avoid this periodic behaviour.

## 7 CIFAR-10 EXPERIMENTS

To compare SGD, signSGD and Adam on less of a toy problem, we ran a large grid search over hyperparameters for training Resnet-20 (He et al., 2016) on the CIFAR-10 dataset (Krizhevsky, 2009). Results are plotted in Figure 2. We evaluate over the hyperparamater 3-space of (initial learning rate, weight decay, momentum), and plot slices to demonstrate the general robustness of each algorithm. We find that, as expected, signSGD and Adam have broadly similar performance. For hyperparameter configurations where SGD is stable, it appears to perform better than Adam and signSGD. But Adam and signSGD appear more robust up to larger learning rates. Full experimental details are given in Appendix A.

## 8 DISCUSSION

First we wish to discuss the connections between signSGD and Adam (Kingma & Ba, 2015). Note that setting the Adam hyperparameters $\beta_1 = \beta_2 = \epsilon = 0$, Adam and signSGD are equivalent. Indeed the authors of the Adam paper suggest that during optimisation the Adam step will commonly look like a binary vector of $\pm 1$ (multiplied by the learning rate) and thus resemble the sign gradient step. If this algorithmic correspondence is valid, then there seems to be a discrepancy between our theoretical results and the empirical good performance of Adam. Our convergence rates suggest that signSGD should be worse than SGD by roughly a factor of dimension $d$. In deep neural network applications $d$ can easily be larger than $10^6$. We suggest a resolution to this proposed discrepancy—there is structure present in deep neural network error surfaces that is not captured by our simplistic theoretical assumptions. We have already discussed in Section 5 how the signSGD bound is improved by a factor $d$ in the case of gradients distributed uniformly across dimensions. It is also reasonable to expect that neural network error surfaces might exhibit only weak coupling across dimensions. To provide intuition for how such an assumption can help improve the dimension scaling of signSGD, note that in the idealised case of total decoupling (the Hessian is everywhere diagonal) then the problem separates into $d$ independent one dimensional problems, so the dimension dependence is lost.

Next, let's talk about saddle points. Though general non-convex functions are littered with local minima, recent work rather characterises successful optimisation as the evasion of a web of saddle points (Dauphin et al., 2014). Current theoretical work focuses either on using noise Levy (2016); Jin et al. (2017) or curvature information (Allen-Zhu, 2017b) to establish bounds on the amount of time needed to escape saddle points. We noted that merely passing the gradient through the sign operation introduces an algorithmic instability close to saddle points, and we wanted to empirically investigate whether this could be enough to escape them. We removed stochasticity from the algorithm to focus purely on the effect of the sign function.

We found that when the objective function was axis aligned, then sign gradient descent without stochasticity (signGD) made progress unhindered by the saddles. We suggest that this is because signGD has a greater ability to 'explore', meaning it typically takes larger steps in regions of small gradient than SGD, and it can take steps almost orthogonal to the true gradient direction. This exploration ability could potentially allow it to break out of subspaces convergent on saddle points without sacrificing its convergence rate—we hypothesise that this may contribute to the often more robust practical performance of algorithms like Rprop and Adam, which bear closer relation to signSGD than SGD. For non axis-aligned objectives, signGD could sometimes get stuck in perfect periodic orbits around saddle points, though we hypothesise that this behaviour may be much less likely for higher dimensional objectives (the testbed function had dimension 10) with non-constant learning rate.

Finally we want to discuss the implications of our results for gradient quantisation schemes. Whilst we do not analyse the multi-machine case of distributed optimisation, we imagine that our results will extend naturally to that setting. In particular our results stand as a proof of concept that we can provide guarantees for *biased* gradient quantisation schemes. Existing quantisation schemes with guarantees require delicate randomisation to ensure unbiasedness. If a scheme as simple as ours can yield provable guarantees on convergence, then there is a hope that exploring further down this avenue can yield new and useful practical quantisation algorithms.

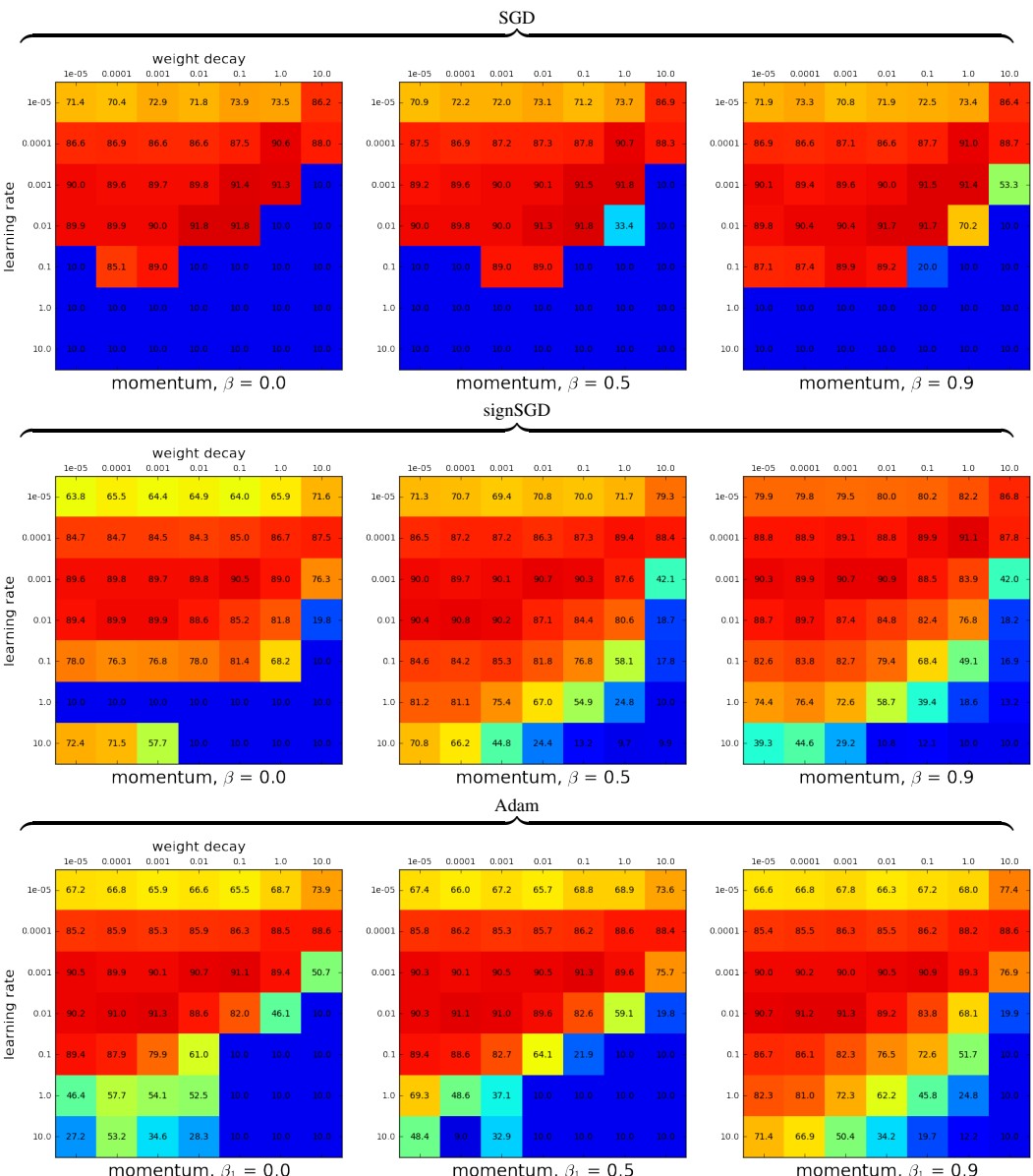

Figure 2: Results for training Resnet-20 (He et al., 2016) on CIFAR-10 (Krizhevsky, 2009) for SGD, signSGD and Adam. We plot test errors over a large grid of initial learning rate, weight decay and momentum combinations. (In signSGD, momentum corresponds to taking the sign of a moving average of gradients—see Appendix A for the detailed experimental setup.) All algorithms at the least get close to the baseline reported in (He et al., 2016) of 91.25%. Note the broad similarity in general shape of the heatmap between Adam and signSGD, supporting a notion of algorithmic similarity. Also note that whilst SGD has a larger region of very high-scoring hyperparameter configurations, signSGD and Adam appear more stable for large learning rates.

## 9 CONCLUSION

We have investigated the theoretical properties of the sign stochastic gradient method (signSGD) as an algorithm for non-convex optimisation. The study was motivated by links that the method has both to deep learning stalwarts like Adam and Rprop, as well as to newer quantisation algorithms that intend to cheapen the cost of gradient communication in distributed machine learning. We have proved non-convex convergence rates for signSGD to first order critical points. Insofar as the rates

can directly be compared, they are of the same order as SGD in terms of number of gradient evaluations, but worse by a linear factor in dimension. SignSGD has the advantage over existing gradient quantisation schemes with provable guarantees, in that it doesn't need to employ randomisation tricks to remove bias from the quantised gradient.

We wish to propose some interesting directions for future work. First our analysis only looks at convergence to first order critical points. Whilst we present preliminary experiments exhibiting success and failure modes of the algorithm around saddle points, a more detailed study attempting to pin down exactly when we can expect signSGD to escape saddle points efficiently would be welcome. This is an interesting direction seeing as existing work always relies on either stochasticity or second order curvature information to avoid saddles. Second the link that signSGD has to both Adam-like algorithms and gradient quantisation schemes is enticing. In future work we intend to investigate whether this connection can be exploited to develop large scale machine learning algorithms that get the best of both worlds in terms of optimisation speed and communication efficiency.

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

## A    EXPERIMENTAL DETAILS

Here we describe the experimental setup for the CIFAR-10 (Krizhevsky, 2009) experiments using the Resnet-20 architecture (He et al., 2016). We tuned over {weight decay, momentum, initial learning rate} for optimisers in {SGD, signSGD, Adam}.

We used our own implementation of each optimisiation algorithm. Adam was implemented as in (Kingma & Ba, 2015) with $\beta_2 = 0.999$ and $\epsilon = 10^{-8}$, and $\beta_1$ was tuned over. For both SGD and signSGD we used a momentum sequence

$$m_{k+1} = \beta m_k + (1 - \beta)\tilde{g}_k \tag{11}$$

and then used the following updates:

$$\text{SGD}: x_{k+1} = x_k - \delta_k m_{k+1} \tag{12}$$
$$\text{signSGD}: x_{k+1} = x_k - \delta_k \text{sign}(m_{k+1}) \tag{13}$$

Weight decay was implemented in the traditional manner of augmenting the objective function with a quadratic penalty.

All other details not mentioned (learning rate schedules, network architecture, data augmentation, etc.) are as in (He et al., 2016). In particular for signSGD we did not use the learning rate or mini-batch schedules as provided by our theory. Code will be released if the paper is accepted.

## B    PROVING THE CONVERGENCE RATE OF THE SIGN GRADIENT METHOD

**Theorem 1** (Non-convex convergence rate of signSGD). *Apply Algorithm 1 under Assumptions 1, 2 and 3. Schedule the learning rate and mini-batch size as*

$$\delta_k = \frac{\delta}{\sqrt{k+1}} \qquad n_k = k + 1 \tag{4}$$

*Let $N_K$ be the cumulative number of stochastic gradient calls up to step $K$, i.e. $N_K = O(K^2)$ Then we have*

$$\mathbb{E}\left[\min_{0 \le k \le K-1} \|g_k\|_1\right]^2 \le \frac{1}{\sqrt{N_{K-2}}}\left[\frac{f_0 - f_*}{\delta} + d(2 + \log(2N_{K-1}))(\sigma + \delta L)\right]^2 \tag{5}$$

*Proof.* Our general strategy will be to show that the expected objective improvement at each step will be good enough to guarantee a convergence rate in expectation. First let's bound the improvement of the objective during a single step of the algorithm for one instantiation of the noise. Note that $\mathbb{I}[.]$ is the indicator function, and $g_{k,i}$ denotes the $i^{th}$ component of the vector $g_k$.

First use Assumption 2, plug in the step from Algorithm 1, and decompose the improvement to expose the stochasticity-induced error:

$$\begin{aligned}
f_{k+1} - f_k &\le g_k^T(x_{k+1} - x_k) + \frac{L}{2}\|x_{k+1} - x_k\|_2^2 \\
&= -\delta_k g_k^T \text{sign}(\bar{g}_k) + \delta_k^2 \frac{L}{2}d \\
&= -\delta_k \|g_k\|_1 + 2\delta_k \sum_{i=1}^{d} |g_{k,i}| \, \mathbb{I}[\text{sign}(\bar{g}_{k,i}) \ne \text{sign}(g_{k,i})] + \delta_k^2 \frac{L}{2}d
\end{aligned}$$

Next we find the expected improvement at time $k+1$ conditioned on the previous iterates.

$$\mathbb{E}[f_{k+1} - f_k | x_k] \le -\delta_k \|g_k\|_1 + 2\delta_k \sum_{i=1}^{d} |g_{k,i}| \, \mathbb{P}[\text{sign}(\bar{g}_{k,i}) \ne \text{sign}(g_{k,i})] + \delta_k^2 \frac{L}{2}d$$

Note that the expected improvement crucially depends on the probability that each component of the sign vector is correct. Intuition suggests that when the magnitude of the gradient $|g_{k,i}|$ is much larger than the typical scale of the noise $\sigma$, then the sign of the stochastic gradient will most likely be correct. Mistakes will typically only be made when $|g_{k,i}|$ is smaller than $\sigma$. We can make this intuition rigorous using Markov's inequality and our variance bound on the noise (Assumption 3).

$$
\begin{aligned}
\mathbb{P}[\text{sign}(\bar{g}_{k,i}) \neq \text{sign}(g_{k,i})] &\leq \mathbb{P}[|\bar{g}_{k,i} - g_{k,i}| \geq |g_{k,i}|] && \text{relaxation} \\
&\leq \frac{\mathbb{E}[|\bar{g}_{k,i} - g_{k,i}|]}{|g_{k,i}|} && \text{Markov's inequality} \\
&\leq \frac{\sqrt{\mathbb{E}[(\bar{g}_{k,i} - g_{k,i})^2]}}{|g_{k,i}|} && \text{Jensen's inequality} \\
&\leq \frac{\sigma_k}{|g_{k,i}|} && \text{Assumption 3}
\end{aligned}
$$

This says explicitly that the probability of the sign being incorrect is controlled by the relative scale of the noise to each component of the gradient magnitude. We denote the noise scale as $\sigma_k$ since it refers to the stochastic gradient with a mini-batch size of $n_k = k + 1$. We can plug this result into the previous expression, take the sum over $i$, and substitute in our learning rate and mini-batch schedules as follows:

$$
\begin{aligned}
\mathbb{E}[f_{k+1} - f_k | x_k] &\leq -\delta_k \|g_k\|_1 + 2\delta_k d\sigma_k + \delta_k^2 \frac{L}{2} d \\
&= -\frac{\delta}{\sqrt{k+1}} \|g_k\|_1 + 2d\frac{\delta\sigma}{k+1} + \frac{\delta^2}{k+1}\frac{L}{2}d \\
&\leq -\frac{\delta}{\sqrt{K}} \|g_k\|_1 + \frac{2\delta d}{k+1}(\sigma + \delta L)
\end{aligned}
$$

In the last line we made some relaxations which will not affect the general scaling of the rate. Now take the expectation over the noise in all previous iterates, and sum over $k$:

$$
\begin{aligned}
f_0 - f^* &\geq f_0 - \mathbb{E}[f_K] && \text{Assumption 1} \\
&= \mathbb{E}\left[\sum_{k=0}^{K-1} f_k - f_{k+1}\right] && \text{telescope} \\
&\geq \mathbb{E}\left[\sum_{k=0}^{K-1} \frac{\delta}{\sqrt{K}} \|g_k\|_1 - \frac{2\delta d}{k+1}(\sigma + \delta L)\right] && \text{previous result} \\
&\geq \mathbb{E}\left[\sum_{k=0}^{K-1} \frac{\delta d}{\sqrt{K}} \|g_k\|_1\right] - 2\delta d(1 + \log K)(\sigma + \delta L) && \text{harmonic sum}
\end{aligned}
$$

We can rearrange this inequality to yield a rate:

$$
\begin{aligned}
\mathbb{E}\left[\min_{0 \leq k \leq K-1} \|g_k\|_1\right] &\leq \mathbb{E}\left[\sum_{k=0}^{K-1} \frac{1}{K} \|g_k\|_1\right] \\
&\leq \frac{1}{\sqrt{K}}\left[\frac{f_0 - f_*}{\delta} + 2d(1 + \log K)(\sigma + \delta L)\right]
\end{aligned}
$$

Since we are growing our mini-batch size, it will take $N_{K-1} = \frac{K(K+1)}{2}$ gradient evaluations to reach step $K-1$. Using that $2N_{K-2} \leq K^2 \leq 2N_{K-1}$ yields the result. For the sake of presentation, we take the final step of squaring the bound, to make it more comparable with the SGD bound.

$\square$

## C   Proving the convergence rate of stochastic gradient descent

**Theorem 2** (Non-convex convergence rate of stochastic gradient descent). *Apply Algorithm 2 under Assumptions 1, 2 and 3. Schedule the learning rate and mini-batch size as*

$$\delta_k = \frac{\delta}{\sqrt{k+1}} \qquad n_k = 1 \qquad (6)$$

*Let $N_K$ be the cumulative number of stochastic gradient calls up to step $K$, i.e. $N_K = K$. Then we have that*

$$\mathbb{E}\left[\min_{0 \le k \le K-1} \|g_k\|_2^2\right] \le \frac{1}{\sqrt{N_K}}\left[\frac{f_0 - f^*}{\delta\left(1 - \frac{\delta L}{2}\right)} + d(1 + \log N_K)\frac{\frac{\delta L}{2}}{1 - \frac{\delta L}{2}}\sigma^2\right] \qquad (7)$$

*Proof.* Consider the objective improvement in a single step, under one instantiation of the noise. Use Assumption 2 followed by the definition of the algorithm.

$$f_{k+1} - f_k \le g_k^T(x_{k+1} - x_k) + \frac{L}{2}\|x_{k+1} - x_k\|_2^2$$

$$= -\delta_k g_k^T \bar{g}_k + \delta_k^2 \frac{L}{2}\|\bar{g}_k\|_2^2$$

Take the expectation conditioned on previous iterates, and decompose the mean squared stochastic gradient into its mean and variance. Note that since $\sigma^2$ is the variance bound for each component, the variance bound for the full vector will be $d\sigma^2$.

$$\mathbb{E}[f_{k+1} - f_k | x_k] \le -\delta_k\|g_k\|_2^2 + \delta_k^2\frac{L}{2}\left(\|g_k\|_2^2 + d\sigma^2\right)$$

Plugging in the learning rate schedule, and using that $\frac{1}{k+1} \le \frac{1}{\sqrt{k+1}}$, we get that

$$\mathbb{E}[f_{k+1} - f_k | x_k] \le -\frac{\delta}{\sqrt{k+1}}\|g_k\|_2^2 + \frac{\delta^2}{k+1}\frac{L}{2}\|g_k\|_2^2 + \frac{\delta^2}{k+1}\frac{L}{2}\sigma^2 d$$

$$\le -\frac{\delta}{\sqrt{k+1}}\|g_k\|_2^2\left(1 - \frac{\delta L}{2}\right) + \frac{\delta^2}{k+1}\frac{L}{2}\sigma^2 d$$

Take the expectation over $x_k$, sum over $k$, and we get that

$$f_0 - f^* \ge f_0 - \mathbb{E}[f_K]$$

$$= \mathbb{E}\left[\sum_{k=0}^{K-1} f_k - f_{k+1}\right]$$

$$\ge \sum_{k=0}^{K-1}\left[\frac{\delta}{\sqrt{k+1}}\mathbb{E}\left[\|g_k\|_2^2\right]\left(1 - \frac{\delta L}{2}\right) - \frac{\delta^2}{k+1}\frac{L}{2}\sigma^2 d\right]$$

$$\ge K\left[\frac{\delta}{\sqrt{K}}\mathbb{E}\left[\min_{0 \le k \le K-1}\|g_k\|_2^2\right]\left(1 - \frac{\delta L}{2}\right)\right] - \sum_{k=0}^{K-1}\left[\frac{\delta^2}{k+1}\frac{L}{2}\sigma^2 d\right]$$

$$\ge \sqrt{K}\delta\,\mathbb{E}\left[\min_{0 \le k \le K-1}\|g_k\|_2^2\right]\left(1 - \frac{\delta L}{2}\right) - (1 + \log K)\delta^2\frac{L}{2}\sigma^2 d$$

And rearranging yields the result.

$$\square$$

