# OpenReview forum: "Convergence rate of sign stochastic gradient descent for non-convex functions"
_ICLR.cc/2018/Conference — Reject_

### Official Review · AnonReviewer1 · 2017-11-23
**Not correct**

**Rating:** 4
**Confidence:** 4

**Review:**

The paper presents convergence rate of a quantized SGD, with biased quantization - simply taking a sign of each element of gradient.

The stated Theorem 1 is incorrect. Even if the stated result was correct, it presents much worse rate for a weaker notion of convergence.

Major flaws:
1. As far as I can see, Theorem 1 should depend on 4th root of N_K, the last (omitted) step from the proof is done incorrectly. This makes it much worse than presented.
2. Even if this was correct, the main point is that this is "only" d times worse - see eq (11). That is enormous difference, particularly in settings where such gradient compression can be relevant. Also, it is lot more worse than just d times:
3. Again in eq (11), you compare different notions of convergence - E[||g||_1]^2 vs. E[||g||_2^2]. In particular, the one for signSGD is the weaker notion - squared L1 norm can be d times bigger again. If this is not the case for some reason, more detailed explanation is needed.

Other than that, the paper contains several attempts at intuitive explanation, which I don't find correct. Inclusion of Assumption 3 would in particular require better justification.

Experiments are also inconclusive, as the plots show convergence to significantly worse accuracy than what the models converged to in original contributions.

---

> ### Author Response · Authors · 2017-12-12
> **We disagree**
>
> Thanks for the feedback, we really appreciate it. We think the "flaws" you mention are actually resulting from some confusion which we will try to clarify here and in the paper.
>
> First of all, the final step of the proof---left implicit---is to square the bound. This gives N^(-1/2) and not N^(-1/4). We will make this explicit to clear up the confusion.
>
> Next, the L_1 norm is indeed larger than L_2 norm. This makes our result stronger! Take the case where L_1^2 = d * L_2^2. Then substitute this into our bound and divide by d on both sides. This improves the dimension dependence of our bound to match SGD. (The intuition here is that when the gradient vector has components uniform in magnitude, then the sign operation preserves direction, and signSGD gets the same dimension dependence as SGD).
>
> We state clearly throughout that there is a gap between our theory which applies to all non-convex functions, and deep network optimisation in practice. One of our contributions is to point out this discrepancy. In particular we suggest that the worse dimension dependence of our bound may not be visible in deep net training because neural network error landscapes have special structure. Non-convex theorists might make use of this observation to design algorithms better suited to neural nets.
>
> We have now replaced assumption 3 (sub-gaussianity) with a new assumption of bounded variance, which is the typical assumption in the SGD literature. We have also run more rigorous experiments where the baselines behave as they do in the original contributions. We will update the draft shortly.
>
> Thanks again for your feedback.

---

> > ### Comment · AnonReviewer1 · 2018-01-02
> > **.**
> >
> > My initial review was perhaps too superficial, I apologize, but the overall feeling holds.
> >
> > Theoretical result is significantly weaker than other existing alternatives, and thus cannot form basis for contribution.
> >
> > It is impossible to draw any conclusions from experiments - on MNIST, you report to converge to ~98.2% accuracy with SGD. The only thing it shows is that you are doing something wrong. The same for CIFAR - you report ~82% accuracy with ResNet18, but original paper shows ~91% with ResNet20. I don't see how can this gap be explained.

---

> > > ### Author Response · Authors · 2018-01-05
> > > **New draft**
> > >
> > > Thanks for looking over our work again.
> > >
> > > Agreed about the experiments. We made poor hyperparameter choices. To rectify this, we ran a large grid search over learning rate, momentum and weight decay, and have put these results in the new draft (Section 7 & Figure 2). The results properly reproduce the baselines.
> > >
> > > To defend our theoretical result as a basis for contribution, we note that there is a huge swell of interest in understanding the theoretical properties of Adam. We claim that the right place to start is understanding the success and failure modes of signSGD, since Adam is closely related but more complicated.
> > >
> > > In the new draft we clarify that for problems with gradients roughly uniformly distributed across dimensions, the dimension dependence of our bound matches SGD.

---

> > > > ### Comment · AnonReviewer1 · 2018-01-10
> > > > **Re**
> > > >
> > > > Thanks, the experiments are indeed an improvement, I have improved my score, but still think this is insufficient. In particular, the result is far worse than for instance Alistarh et al., I recommend reshaping this work as rather experimental evaluation in the future.

---

> > > > > ### Author Response · Authors · 2018-01-17
> > > > > **Alistarh et al. can have worse dimension dependence than our result**
> > > > >
> > > > > In what sense is the result far worse than Alistarh et al.?
> > > > >
> > > > > We have now validated empirically that for resnet-20 on cifar-10, the squared gradient 1-norm dominates the squared gradient 2-norm by a factor O(d). Also the stochastic gradient variance is O(d).
> > > > >
> > > > > The closest thing to our result in Alistarh et al. is Theorem 3.5, setting s=1 for quantisation levels of -1, +1, 0. Note that B in their theorem is O(d). Therefore the right hand side of their bound is of order d^1.5, whereas ours is of order d.
> > > > >
> > > > > [Note that in their notation, d=n. Note also that there is a typo in their Theorem 3.5, it should depend on f(x) - f* and not sqrt(f(x) - f*)]

---

### Official Review · AnonReviewer2 · 2017-11-25
**[UPDATED] Would rate more confidently, if stronger numerical experiments are present :) and Assumption 3 is more explained and defended**

**Rating:** 4
**Confidence:** 4

**Review:**

UPDATED REVIEW:

I have checked all the reviews, also checked the most recent version.
I like the new experiments, but I am not impressed much with them to increase my score. The assumption about the variance is fixing my concern, but as you have pointed out, it is a bit more tricky :) I would really suggest you work on the paper a bit more and re-submit it.

--------------------------------------------------------------------
In this paper, authors provided a convergence analysis of Sign SGD algorithm for non-covex case.
The crucial assumption for the proof was Assumption 3, otherwise, the proof technique is following a standard path in non-convex optimization.

In general, the paper is written nicely, easy to follow.

==============================================
"The major issue":
Why Assumption 3 can be problematic in practice is given below:
Let us assume just a convex case and assume we have just 2 kids of function in 2D:  f_1(x) = 0.5 x_1^2 and f_2(x) = 0.5 x_2^2.
Then define the function f(x) = E [ f_i(x)  ].   where $i =1$  with prob 0.5 and $i=2$ with probability 0.5.
We have that   g(x) = 0.5 [ x_1, x_2 ]^T.
Let us choose $i=1$ and choose $x = [a,a]^T$, where $a$ is some parameter.

Then (4) says, that there has to exist a $\sigma$ such that
P [   | \bar g_i(x) - g_i(x) | > t ] \leq 2 exp( - t^2 / 2\sigma^2).  forall "x".

plugging our function inside it should be true that

P [   | [ B ] - 0.5 a | > t ] \leq 2 exp( - t^2 / 2\sigma^2).  forall "x".
where B is a random variable which has value "a" with probability 0.5 and value "0" with probability 0.5.

If we choose $t = 0.1a$ then we have that it has to be true that

1 = P [   | [ B ] - 0.5 a | > 0.1a ] \leq 2 exp( - 0.01 a^2 / 2\sigma^2)   ---->  0 as $a \to \infty$.

Hence, even in this simple example, one can show that this assumption is violated unless $\sigma = \infty$.

One way to ho improve this is to put more assumption + maybe put some projection into a compact set?
==============================================

Hence, I think the theory should be improved.

In terms of experiments, I like the discussion about escaping saddle points, it is indeed a good discussion. However, it would be nicer to have more numerical experiments.
One thing I am also struggling is the "advantage" of using signSGD: one saves on communication (instead of sending 4*8 bits per dimension, one just send only 1 bit, however, one needs "d"times more iterations, hence, the theory shows that it is much worse then SGD (see (11) ).

---

> ### Author Response · Authors · 2017-12-12
> **We changed assumption 3 to bounded variance**
>
> Thanks for the review! We really appreciate it, and the example you give is great. It boils down to a construction of a finite sum problem where the stochastic gradient variance diverges when x tends to infinity.
>
> Since submitting, we have modified the proof to swap Assumption 3 for an assumption of bounded variance. Though bounded variance is the standard assumption in the SGD literature, it still fails under your example. Indeed the problem can be fixed by projecting to a compact set as you say, but we prefer to keep the assumption of bounded variance since it makes our work directly comparable with the existing literature.
>
> In practice signSGD is immensely useful, since it converges fast for deep nets, and also uses quantised gradients. We agree that there is a gap between our theory which uses standard assumptions and applies to all non-convex functions, and practice where we test on deep neural networks. Drawing attention to this gap may be one of the main contributions of our paper---we imply that if non-convex theorists want to have more impact on deep learning practice, we need to adopt assumptions that better capture the geometry of deep neural net objective functions. (Another possibility is just that our bound is not tight, and we are working on constructing a lower bound to check this.)
>
> We will update the paper shortly with the *new* assumption 3 of bounded variance, and more rigorous experiments. Thank you for the suggestions :)

---

> ### Author Response · Authors · 2018-01-05
> **Updated draft**
>
> Dear Reviewer,
>
> We have updated our draft:
>
> 1) we change assumption 3 for a simpler variance bound
> 2) we include a more extensive experimental study
> 3) we clarify that for problems with gradient distributed roughly uniform across dimensions, signSGD acquires the same dimension dependence as SGD (section 5)
>
> Thank you for your feedback :)

---

### Official Review · AnonReviewer3 · 2017-11-27
**Preliminary work that requires further investigation**

**Rating:** 5
**Confidence:** 5

**Review:**

Dear Authors,
After reading the revised version I still believe that the assumption about the gradients + their variances to be distributed equivalently among all direction is very non-realistic, also for the case of deep learning applications.

I think that the direction you are taking is very interesting, yet the theoretical work is still too preliminary and I believe that further investigation should be made in order to make a more complete manuscript.

The additional experiments are nice.  I therefore raised my score by a bit.


$$$$$$$$$$$$$$$$$$$$$$$$$$$$$$$$$$$$$$$$$$$$$$$$$$$$$$$$$$$$$$$$$$$$$$$$
 The paper explores SignGD --- an algorithm that uses the sign of the gradients instead of actual gradients for training deep models. The authors provide some guarantees regarding the convergence of SignGD to local minima in the stochastic optimization setting, and later compare SignSG to GD in two deep learning tasks.

Exploring signSGD is an important and interesting line of research, and this paper provides some preliminary result in this direction.
However, in my view, this work is too preliminary and not ready for publish. This is since the authors do not illustrate any clear benefits of signSGD over SGD neither in theory nor in practice. I elaborate on this below:

-The theory part shows that under some conditions, signGD  finds a local minima. Yet, as the authors themselves
mention, the dependence on the dimension is much worse compared to SGD.
Moreover, the authors do not mention that if the noise variance does not scale with the dimension (as is often the case), then the convergence of SGD will not depend on the dimension, while it seems that the convergence of signGD will still depend on the dimension.

-The experiments are nice as a preliminary investigation, but not enough in order to illustrate the benefits of signSGD over SGD. In order to do so, the authors should make a more extensive experimental study.

---

> ### Author Response · Authors · 2017-12-12
> **signSGD is good in practice due to fast empirical convergence, and quantised gradients**
>
> Thanks for reviewing our paper---we really appreciate the feedback! We're very interested in your comment about the dimension dependence of the noise variance---would you be able to point us to an example where it does not depend on dimension?
>
> We view the contribution of our work as twofold. First at the empirical level, we show that signSGD (a method that 1-bit quantises gradients) has empirical convergence properties in deep learning tasks that rival SGD. Therefore we have shown that in practice the method is immensely useful for distributed optimisation, since it converges fast AND has cheap gradient communication across machines. Our method is much simpler than other quantised gradient schemes that take pains to ensure the quantisation scheme is unbiased. We show that in practice unbiasedness is not necessary. Indeed we have now run more rigorous experiments to demonstrate this, and we will update the draft shortly.
>
> Second on the theoretical level, we put signSGD on the same theoretical footing as SGD, for non-convex functions. Until now there was no non-convex theory of this method. Our work is the first step. We clearly state that signSGD has worse dimension dependence than SGD, but this holds for all non-convex functions. Our assumptions are typical for non-convex theory papers. The surprising observation is that in theory the method is worse, but in practice for neural networks it performs the same, therefore we suggest that there may be special structure in neural network error landscapes, which is not captured by the typical assumptions of non-convex theory work. We are working on constructing a lower bound to check the alternative hypothesis that our bound is just not tight.

---

> > ### Comment · AnonReviewer3 · 2017-12-12
> > **Example**
> >
> > Dear Authors,
> >
> > Scenarios with dimension independent variance often arise in text classification.
> > Where each word in a dictionary appears with probability p_i, and p_i is a heavy tailed distribution (e.g. geometric distribution)
> > In such scenarios, it can be shown that the total variance is dimension independent.
> > For a detailed description of this setup you can look in McMahan and Streeter 2010, see Section 1.2  https://arxiv.org/pdf/1002.4908.pdf.

---

> > > ### Author Response · Authors · 2017-12-16
> > > **Thanks for the example**
> > >
> > > Thank you for the quick reply, and for the reference :)
> > >
> > > We want to point out that:
> > >
> > > 1. our bound will also benefit from dimension independent variance in the gradients
> > >
> > > 2. our bound is on the L1 norm of the gradient. For problems where the gradient is typically uniform in magnitude across dimensions, then the square L1 norm is roughly d times larger than square L2 norm. Therefore our bound acquires the same dimension dependence as the SGD bound in this setting.
> > >
> > > 3. the reference you give is very interesting, but it is not clear how relevant it is for deep networks. In particular, deep networks can suffer from problems like exploding gradients. A priori, it seems that exploding gradient type phenomena should at least lead to gradient variances that depend on network depth.

---

> > > ### Author Response · Authors · 2018-01-05
> > > **Updated draft**
> > >
> > > Dear Reviewer,
> > >
> > > We have updated our draft with:
> > > 1) a more extensive experimental study (Section 7)
> > > 2) a simpler assumption on the stochastic gradient noise model (Assumption 3)
> > > 3) a simple condition under which dimension dependence of our bound matches SGD (Section 5)
> > >
> > > Thanks for your feedback throughout this process :)

---

> > > > ### Comment · AnonReviewer3 · 2018-01-10
> > > > **New Draft**
> > > >
> > > > Dear Authors,
> > > >
> > > > After reading the revised version I still believe that the assumption about the gradients + their variances to be distributed equivalently among all direction is very non-realistic, also for the case of deep learning applications.
> > > >
> > > > I think that the direction you are taking is very interesting, yet the theoretical work is still too preliminary and I believe that further investigation should be made in order to make a more complete manuscript.
> > > >
> > > > The additional experiments are nice.  I therefore raised my score by a bit.

---

> > > > > ### Author Response · Authors · 2018-01-17
> > > > > **Validated assumptions about gradients empirically.**
> > > > >
> > > > > Dear Reviewer,
> > > > >
> > > > > For your interest and for the sake of posterity, we have run experiments to test our assertions about gradient statistics for Resnet-20 architecture, Cifar-10 dataset. We find that our assertions do hold up.
> > > > >
> > > > > In particular, we find that
> > > > > (i) squared 1-norm of gradient dominates the squared 2-norm by a factor of order d throughout training
> > > > > (ii) the stochastic gradient variance is also of order d throughout training

---

### Author Response · Authors · 2018-01-05
**Updated draft of paper + Relevant parallel work**

Dear Reviewers and Area Chair,

There is a relevant parallel work submitted to ICLR called "Dissecting Adam: The Sign, Magnitude and Variance of Stochastic Gradients" (https://openreview.net/forum?id=S1EwLkW0W)
-- we became aware of this work only after submission
-- the reviewers of "Dissecting Adam" raised the lack of non-convex theory as an issue with their analysis of signSGD. Our paper addresses this point.

We uploaded a new version of our paper with main changes as follows:
-- changed the stochastic gradient noise model from sub-Gaussian to bounded variance (assumption 3)
-- replaced the CIFAR-10 experiments with more robust ones---a large sweep over hyperparameter space (section 7)
-- clarified that when gradients are uniformly distributed across dimensions, the signSGD bound acquires same dimension dependence as SGD bound (section 5)

Thanks!

---

> ### Comment · AnonReviewer1 · 2018-01-10
> **Another Adam link**
>
> Along the lines of contrasting with other ICLR submissions, have a look at this one too, which seems to work against some of your claims below.
> https://openreview.net/forum?id=ryQu7f-RZ

---

> > ### Author Response · Authors · 2018-02-16
> > **Re: Another Adam link**
> >
> > Please see my reply to the Program Chairs above. Thank you
> >
> > --Jeremy

---

### Decision · Program_Chairs · 2018-01-29
**ICLR 2018 Conference Acceptance Decision**

**Decision:**

Reject

**Comment:**

Dear authors,

After carefully reading the reviews, the rebuttal, and going through the paper, I regret to inform you that this paper does not meet the requirements for publication at ICLR.

While the variance analysis is definitely of interest, the reality of the algorithm does not match the claims. The theoretical rate is worse than that of SG but this could be an artefact of the analysis. Sadly, the experimental setup lacks in several ways:
- It is not yet clear whether escaping the saddle points is really an issue in deep learning as the loss function is still poorly understood.
- This analysis is done in the noiseless setting despite your argument being based around the variance of the gradients.
- You report the test error on CIFAR-10. While interesting and required for an ML paper, you introduce an optimization algorithm and so the quantity that matters the most is the speed at which you achieve a given training accuracy. Also, your table lists the value of the test accuracy rather than the speed of increase. Thus, you test the generalization ability of your algorithm while making claims about the optimization performance.

---

> ### Author Response · Authors · 2018-02-16
> **Reply**
>
> Dear Program Chairs and anonReviewers, we appreciate your healthy scepticism.
>
> --For situations where gradients are dense, the signSGD rate matches SGD. We now have experiments to back this up.
> --There was no convincing theory of adaptive gradient methods (e.g. Adam) until now.
> --This competing work https://openreview.net/forum?id=ryQu7f-RZ makes use of bimodal noise distributions to show situations where Adam can fail.
> --We show that such unfriendly noise distributions are excluded by choosing a large enough mini-batch size. Therefore other works characterise failure modes of adaptive methods which may not arise in practice. We characterise success modes which do arise in practice.
>
> Again, thank you for the constructive feedback. Our arxiv version contains the missing experiments you mentioned. https://arxiv.org/abs/1802.04434
>
> Our theoretical work also suggests an elegant algorithm for gradient compression, called majority vote.
>
> --Jeremy Bernstein